# Decomposing spontaneous sign language into elementary movements: A principal component analysis-based approach

**Félix Bigand**[1]*, **Elise Prigent**[1], **Bastien Berret**[2], **Annelies Braffort**[1]

**1** Université Paris-Saclay, CNRS, LISN, Orsay, France, **2** Université Paris-Saclay, CIAMS, Institut Universitaire de France, Orsay, France

* felix.bigand@universite-paris-saclay.fr

## Abstract

Sign Language (SL) is a continuous and complex stream of multiple body movement features. That raises the challenging issue of providing efficient computational models for the description and analysis of these movements. In the present paper, we used Principal Component Analysis (PCA) to decompose SL motion into elementary movements called principal movements (PMs). PCA was applied to the upper-body motion capture data of six different signers freely producing discourses in French Sign Language. Common PMs were extracted from the whole dataset containing all signers, while individual PMs were extracted separately from the data of individual signers. This study provides three main findings: (1) although the data were not synchronized in time across signers and discourses, the first eight common PMs contained 94.6% of the variance of the movements; (2) the number of PMs that represented 94.6% of the variance was nearly the same for individual as for common PMs; (3) the PM subspaces were highly similar across signers. These results suggest that upper-body motion in unconstrained continuous SL discourses can be described through the dynamic combination of a reduced number of elementary movements. This opens up promising perspectives toward providing efficient automatic SL processing tools based on heavy mocap datasets, in particular for automatic recognition and generation.

## Introduction

Sign Languages (SLs) are the first languages of 70 million deaf people in the world [1]. Yet, deaf SL users face many communication barriers. In particular, the vast majority of automatic communication tools are not compatible with SL content, but only with spoken or written one. Developing successful tools for automatic SL processing (i.e., SL automatic recognition, generation and translation) would allow breaking down these barriers. For that aim, further insights must be gained into multiple disciplines, in particular motion science. Indeed, beyond the sparsity of research and developments conducted in SL compared to spoken languages, the automatic processing of SL is challenging because of the intrinsic complexity of SL movements. For instance, SL involves multiple motion features from various body parts, such as

ORTOLANG website (https://www.ortolang.fr/market/corpora/mocap1/).

**Funding:** This work has been funded by the Bpifrance (https://www.bpifrance.fr/) investment project "Grands defis du numerique", as part of the ROSETTA project (RObot for Subtitling and intElligent adapTed TranslAtion). The funders had no role in study design, data collection and analysis, decision to publish, or preparation of the manuscript.

**Competing interests:** The authors have declared that no competing interests exist.

movements of the torso, arms, hands and fingers as well as facial expressions. These movements involve coordinating many biomechanical degrees of freedom (DOFs) [2, 3]. Furthermore, the spatial and temporal coordination of SL gestures is driven by a complex linguistic system, whose modeling does not yet meet with a broad consensus among linguists. This raises the question of determining how the motor system actually controls such complex movements and how computational models of SL motion could be improved by exploiting these control strategies.

Indeed, despite the complexity of SL motion, SL users have no difficulty engaging in various SL conversations throughout the day. More generally, the ease with which humans fluidly execute movements has questioned how the motor system could coordinate the many DOFs and master such a high-dimensional space of postures [4]. One hypothesis is that the multiple DOFs are controlled within a subspace of lower dimension than the available number of DOFs [4–6]. Instead of processing the multiple DOFs individually, the motor system controls a reduced set of compositional elements called synergies (i.e., patterns of muscle activations) [7–9]. These synergies are combined and, depending on their respective weights, are used to generate different movements.

With the progress of motion capture (mocap) systems, converging evidence has been provided across various movement contexts that motion datasets could be properly defined using a reduced number of synergies. For instance, Principal Component Analysis (PCA) has been shown to be effective in extracting synergies, by decomposing motion data into uncorrelated principal movements (PMs). Troje [10] first used PCA to disentangle the motion patterns of human gait. Similarly to "eigenfaces" [11] or "eigenvoices" [12], the whole movement of the walker was decomposed into simpler one-directional PMs (i.e., time series of "eigenpostures"), which maximized the variance in the original motion. The first PMs accounted for most of the variance in the movements and can be interpreted as the kinematic elements (i.e., synergies) recruited by the motor system to organize the movement [13]. PCA has then been successfully applied to various common movements, such as walking or running [14–17], but also in more complex sport contexts, such as skiing [18], karate [19] or diving [20].

In most of the studies mentioned above, PM decomposition has allowed gaining insights into the coordinative structure of complex movements and into the underlying mechanisms of motor control. This data-driven technique has also allowed shedding light on specific motor mechanisms related to gesture expertise. In Federolf et al. [18], PMs were projected back onto the original 3D space and were visualized. This allowed interpreting and comparing the skiing movements of athletes in terms of distinct PMs, such as lateral body inclination, flexion-extension of the legs or rotation of the skis. The same method has been successfully applied to other sports, such as karate [19] or diving [20]. In juggling, individual differences due to experience have also been found in specific PMs [21]. Beyond expertise, such differences in PMs have allowed determining specific biomechanical mechanisms due to impairments, such as knee osteoarthritis [14] or cerebral palsy [15, 22]. Several studies have also used this technique to investigate human posture control [23–27], notably to detect mechanisms due to perturbations of the postural control system, such as head shaking [28]. Additionally, Haid et al. [26] have reported age effects in postural control characterized by control differences in specific PMs. In the artistic domain, Tits et al. [29] have showed that the finger gestures of pianists can be decomposed into eight PMs and that the complexity of the decomposition was a function of the expertise of the pianists.

To the authors' knowledge, such a holistic evaluation of the upper-body movements in spontaneous SL discourses has not been proposed yet. Some studies investigating hand synergies have provided insights into the movements of the dominant hand of signers during the production of highly constrained isolated ASL signs (i.e., shaping the hand into static letters of

the alphabet, or numbers) [30–33]. PCA application was successful, which suggests that hand control of ASL letters occupies a space of low dimension. These results support the potential contribution of PM decomposition to automatic SL processing, as it allows models to use a reduced number of dimensions and thus simplifies automatic tasks such as recognition, as previously shown for ASL letters [30]. However, producing ASL letters or numbers with the hand is only a reduced part of particular ASL signs. Moreover, SL productions made in isolation hardly provide complete descriptions of how SL is used by signers in real-life conditions, even for more complex signs than letters or numbers. For instance, it has been demonstrated that spontaneous SL mocap recordings could reveal faster movements than isolated signs [34–36]. Furthermore, SL motion involves the coordination of far more body parts than the dominant hand, including the other hand, but also torso, head, shoulders and arms.

Investigating PM decomposition for the description of spontaneous upper-body SL movements thus presents a two-fold interest. First, it may provide unexpected fundamental insights into how the complex movements of SL are structured and into the motor control strategies used to produce such complex motion. These analyses could interestingly complement prior studies made on a linguistically limited set of hand gestures [30–33]. Furthermore, it could be used to improve technological tools dedicated to SL, notably as it allows for substantial dimensionality reduction. Indeed, PCA has been successfully applied to human full-body movements for machine learning purposes, such as automatic prediction of gender [10], identity [37], mental state [38] or expertise level [19, 20], which calls for further investigations using PMs as relevant inputs of automatic SL models. For instance, dense mocap datasets could be decomposed using PCA before more complex machine learning procedures, notably considering the crucial role of proper pre-processing steps in improving the performance of deep neural networks for automatic SL recognition [39]. Again, such recognition models could complement the prior work made on the automatic classification of ASL alphabet letters using PCA [30]. Furthermore, taking advantage of the inter-segment coordination with PMs could open up promising perspectives for automatic SL generation via virtual signers (or signing avatars), as previously shown for robot control problems [5], such as control of artificial hands [7, 40, 41]. For instance, animation models could use dense mocap datasets to produce realistic movements by processing only a reduced subset of PMs while keeping most of the information about the original movements.

In the present study, we used PCA in order to determine the extent to which upper-body motion of French Sign Language (LSF) could be described within a low number of PMs. Moreover, we aimed to quantify whether these PMs were shared across various signers in spontaneous discourses. For that aim, (1) common PMs were extracted from a mocap dataset containing spontaneous LSF motion of six signers; (2) individual PMs were extracted from the separate data of each signer; (3) the consistency of the principal movements across signers was assessed.

## Materials and methods

### Motion capture corpus

The data used in the present analyses were taken from a previously reported study [42]. In brief, each of six deaf native and fluent signers (3/3 males/females) had freely described the content of 25 pictures using French Sign Language (LSF). The signers had been selected so that they were all deaf, fluent in LSF and working in professions that require them to be comfortable expressing themselves in front of a camera (e.g., teacher, journalist, story-teller, translator). They all gave informed written consent (translated into LSF for better accessibility) before the experiment, in accordance with the ethical standards of the Declaration of Helsinki.

Using a motion capture system equipped with 10 cameras (Optitrack S250e), the data consisted of the upper-body movements recorded at 250 fps, in three dimensions. Further details (e.g., picture content, type of SL discourse, mocap equipment) are available from the original mocap corpus [43, 44]. From the 25 mocap recordings, only 24 were taken into account in the present study, as one of them was not available for one signer. Moreover, from the 23 original body markers available for all signers, we used 21 markers that optimally describe the major joints of the body. We chose to represent each elbow using one marker instead of two, as the second marker did not add substantial information and removing it eased the visualization of stick figures. As shown in Fig 1, the markers were (L = left, R = right, F = front, B = back): (1) pelvis, (2) stomach, (3) T10 kidneys, (4) sternum, (5) C7 thyroic gland, (6) LB head, (7) LF head, (8) RB head, (9) RF head, (10) L shoulder, (11) L elbow, (12) LB wrist, (13) LF wrist, (14) LB hand, (15) LF hand, (16) R shoulder, (17) R elbow, (18) RB wrist, (19) RF wrist, (20) RB hand, (21) RF hand.

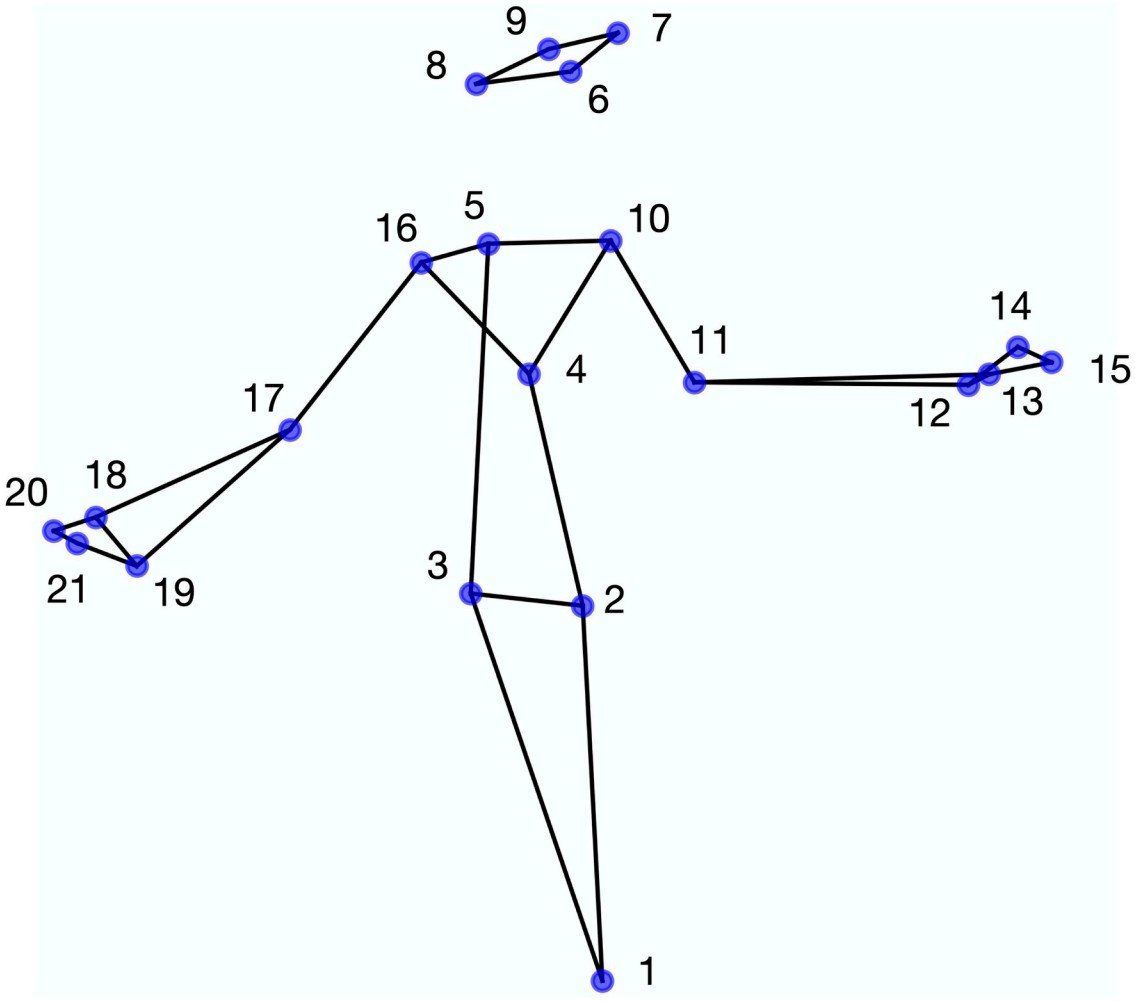

**Fig 1. The 21 upper-body markers taken from the motion capture corpus.**

## Data pre-processing

All the data processing was conducted in Python (Python Software Foundation https://www.python.org/) using custom code, which is publicly available at the following GitHub repository: https://github.com/felixbgd/SL-PMs_Bigandetal_2021.

The 3D positions of all markers were defined in reference to the pelvis (used as the origin). From each of the 24 original recordings, one mocap recording unit with the duration of 5-second was extracted from the beginning of the utterance, irrespective of the semantic content. Each mocap recording unit was thus related to a different SL utterance. This resulted in 24 mocap examples per signer, of 5-second duration each.

The movements of each individual signer (i.e., the concatenation of their 24 mocap examples) were described in a matrix containing 30,000 posture vectors (rows) defined by the 3D coordinates of the 21 markers (columns) at each time frame $t$:

$$\overrightarrow{p(t)} = [x_1(t), y_1(t), z_1(t), \ldots, x_{21}(t), y_{21}(t), z_{21}(t)] \tag{1}$$

A two-step normalization was then applied to these data, in order to allow extracting common PMs across different signers while reducing the effects of anthropometric differences, as defined in prior work [23]. First, the mean posture of each signer was computed over the 24 mocap examples (Eq 2). The mean posture was then subtracted from the posture vectors for each signer (Eq 3), in order to capture the variance caused by postural movements (i.e., deviations from the mean) rather than by differences in the mean postures:

$$\overrightarrow{\overline{p_{signer}}} = [\overline{x_1}, \overline{y_1}, \overline{z_1}, \ldots, \overline{x_{21}}, \overline{y_{21}}, \overline{z_{21}}] \tag{2}$$

where $\overline{x} = mean_t(x(t))$.

$$\overrightarrow{p_{cent}(t)} = \overrightarrow{p(t)} - \overrightarrow{\overline{p_{signer}}} \tag{3}$$

where $\overrightarrow{p_{cent}(t)}$ is the centered posture vector.

Furthermore, the centered posture vectors were normalized to their mean Euclidean norm, in order to ensure an equal contribution by each signer to the variance of the combined matrix. The Euclidean norm was defined as follows:

$$d_{signer}(t) = \|\overrightarrow{p_{cent}}\|_2(t) = \sqrt{x_{cent,1}(t)^2 + \ldots + z_{cent,21}(t)^2} \tag{4}$$

Then, centered posture vectors were divided by the mean of the Euclidean norm. The normalized vectors thus had the following form:

$$\overrightarrow{p_{norm}(t)} = \frac{1}{d_{signer}} \overrightarrow{p_{cent}(t)} = \frac{1}{d_{signer}} (\overrightarrow{p(t)} - \overrightarrow{\overline{p_{signer}}}) \tag{5}$$

In order to investigate common PMs across individuals, the normalized $30,000 \times 57$-posture matrices of the six signers were concatenated into a $180,000 \times 57$-matrix.

## Extraction of principal movements

PMs were extracted by applying PCA to the normalized posture matrix. PCA was performed using singular value decomposition and produced principal components (PCs) (or eigenvectors) and their respective eigenvalues. The normalized eigenvalues indicated the percentage of variance explained by the related PCs. Each posture $\overrightarrow{p(t)}$ could then be reconstructed using a linear combination of the PCs. The PMs were then characterized by projecting separately each

specific PC back onto the original 3D space (Eq 6). PMs were resynthesized using stick figures. This allowed visualizing the PMs and comparing the motion patterns they described.

$$\overrightarrow{PM_i(t)} = \overrightarrow{p_{signer}} + \overline{d_{signer}}\, w_i(t)\overrightarrow{PC_i} \tag{6}$$

where $\overrightarrow{PM_i(t)}$ is the vector that describes the movements of the $i^{th}$ PM. $w_i(t)$ is the projection of the normalized posture vector $\overrightarrow{p_{norm}(t)}$ onto the $i^{th}$ PC-vector, $\overrightarrow{PC_i}$.

In order to ensure that the extracted PMs reflect actual, volitional, movements rather than noise, their frequency content was estimated. For that aim, the Power Spectral Density (PSD) of the $w_i(t)$ was computed using the Welch method [45]. This analysis revealed that the highest power resided in frequencies below 3 Hz, but visible power was still found in the frequency range between 3 to 6 Hz (see S1 Fig). Therefore, the $w_i(t)$ were low-pass filtered using a 4th-order Butterworth Filter with a cut-off frequency of 6 Hz. The effect of noise was thus deemed to have been sufficiently reduced.

## Statistical analyses

This study aimed to assess the extent to which upper-body motion in spontaneous SL could be described within low-dimensional subspaces shared by different signers. Therefore, a leave-one-out cross-validation was conducted to evaluate the vulnerability of the PMs to signer changes in the input (i.e., a signer was added or left out before applying the PCA). On average, the first eight PMs were found to be robust to signer changes (i.e., the PC-vector did not change its orientation in posture space by more than 15° when a signer was left out) [26] and explained around 94.6% of the total variance. They were therefore included in the further analyses. Yet, note that PM5 and PM6 were slightly less robust when Signer 5 was left out (i.e., for this specific signer change, the $5^{th}$ and $6^{th}$ PC-vectors changed their orientations by 21.9° and 20.6°, respectively), which motivated further statistical analyses assessing the extent to which the PMs could be common across different signers.

To address this problem, we further extracted individual PMs (i.e., from the 30, 000 × 57-matrices of each signer separately) and assessed their similarity to the common PMs (i.e., extracted from the 180, 000 × 57-matrix combining the six signers). This analysis notably aimed to investigate whether the PMs could be similar across signers although ranked in a different order by the PCA, notably due to differences in the discourse, which was not constrained and thus could activate PMs differently across signers. The similarities between the common and individual PMs were assessed using cosine similarity [46, 47]. This analysis was conducted on the normalized PMs, in order to compare postural movements beyond anthropometric differences. The 30, 000 × 57-posture matrix of each normalized PM was reshaped into a vector of length 1, 710, 000. Then, the cosine similarity (*sim*) between two PMs was computed as follows:

$$sim(\overrightarrow{PM_i}, \overrightarrow{PM_j}) = \frac{\sum_{k=1}^{N} PM_{i,k}PM_{j,k}}{\sqrt{\sum_{k=1}^{N} PM_{i,k}^2}\sqrt{\sum_{k=1}^{N} PM_{j,k}^2}} \tag{7}$$

where $\overrightarrow{PM_i}$ and $\overrightarrow{PM_j}$ are the 1, 710, 000-vectors of the two PMs to be compared and $N$ is the length of both vectors ($N$ = 1, 710, 000).

Finally, beyond comparing individual and common PMs one-by-one as in the analysis above, we further aimed to compare the PM subspaces specific to each signer. Therefore, we evaluated the extent to which the individual PM subspace of one signer could account for the variance in the movements of the other signers, by computing the cross-projection similarity

[31, 48]. For that aim, the movements of one signer were projected onto their first N PC-vectors and the cumulative amount of variance of these projections was computed ($V_1$). Then, the movements of this first signer were projected onto the first N PC-vectors of a second signer and, similarly, the cumulative amount of variance was computed ($V_2$). Finally, we calculated the ratio $V_2/V_1$, which quantified the extent to which the second subspace was similar to the first. The closer this ratio is to 1, the higher the similarity is. As this measure is not symmetric (i.e., its value would not necessarily be equivalent if we projected the movements of the second signer onto the N PC-vectors of the first one), the cross-projection similarity between the subspaces of two signers was then obtained as the mean of the ratios computed in both directions.

## Results

### Structure of common principal movements

Common PMs were computed from the mocap dataset containing the 24 examples of the six signers. As shown in Fig 2, most of the overall variance was explained by the first eight common PMs. Combined, the first eight common PMs explained 94.6% of the cumulative variance.

The first eight common PMs (Videos 1 to 8 in S1 Dataset) are shown in Fig 3 and are described in details in Table 1. In summary, the first eight PMs were mainly defined as motion patterns visible in the frontal and sagittal planes. PM1 to PM4 quantified movements of the two hands along the vertical, anteroposterior and mediolateral axes, as well as upper-body rotation around the vertical axis. PM5 and PM6 quantified specific joint movements, such as arm internal rotation or elbow flexion. Higher-order PMs (PM7 and PM8) extracted finer movements, such as flexion of the wrists or shoulder abduction. Moreover, PM8 reported covarying movements between the head, the torso and the arms of the signers. For instance, a low negative weighting of PM8 was related to high flexion of the head, anteroposterior inclination of the torso, and high abduction of both shoulders.

### Similarities in principal movements across signers

Individual PMs (i.e., specific to each signer) were computed from the mocap data of individual signers. As shown in Fig 4, most of the overall variance was explained by the first eight individual PMs. Combined, the first eight individual PMs explained 95.7% (SD = 0.6%) of the cumulative variance. By comparison, the first eight common PMs explained 94.6%, and the first seven individual PMs explained 94.3% (SD = 0.9%). Therefore, although the common dataset contained a wide variety of unsynchronized movements performed by six different signers, the common PMs explained a similar amount of the movements variance, compared with PMs computed separately for each signer.

Visual animations of the first eight individual PMs seemed highly similar to the common PMs for most signers, although they may sometimes be ranked in a different order. This observation was further confirmed by the cosine similarity measures, as show in Fig 5. For instance for Signer 4 (Videos 9 to 16 in S1 Dataset), all the first seven PMs reported high levels of similarity ($sim > 0.7$ [47]) and PM8 reported a level of similarity near the latter threshold ($sim = 0.69$). All the PMs of this signer were similar to common PMs of the same order (i.e., individual $i^{th}$ PM is similar to common $i^{th}$ PM), except for PM3 and PM4, which were ranked in reverse order. Of the total number of 48 individual PMs across signers, 25 PMs reported a similarity level above 0.7, and 36 reported a similarity level above 0.5. The amount of similarity could vary depending on the signer. For instance, although some signers also presented consequent levels of similarity (e.g., Signer 3, where $sim > 0.7$ for five PMs, and $sim > 0.5$ for the

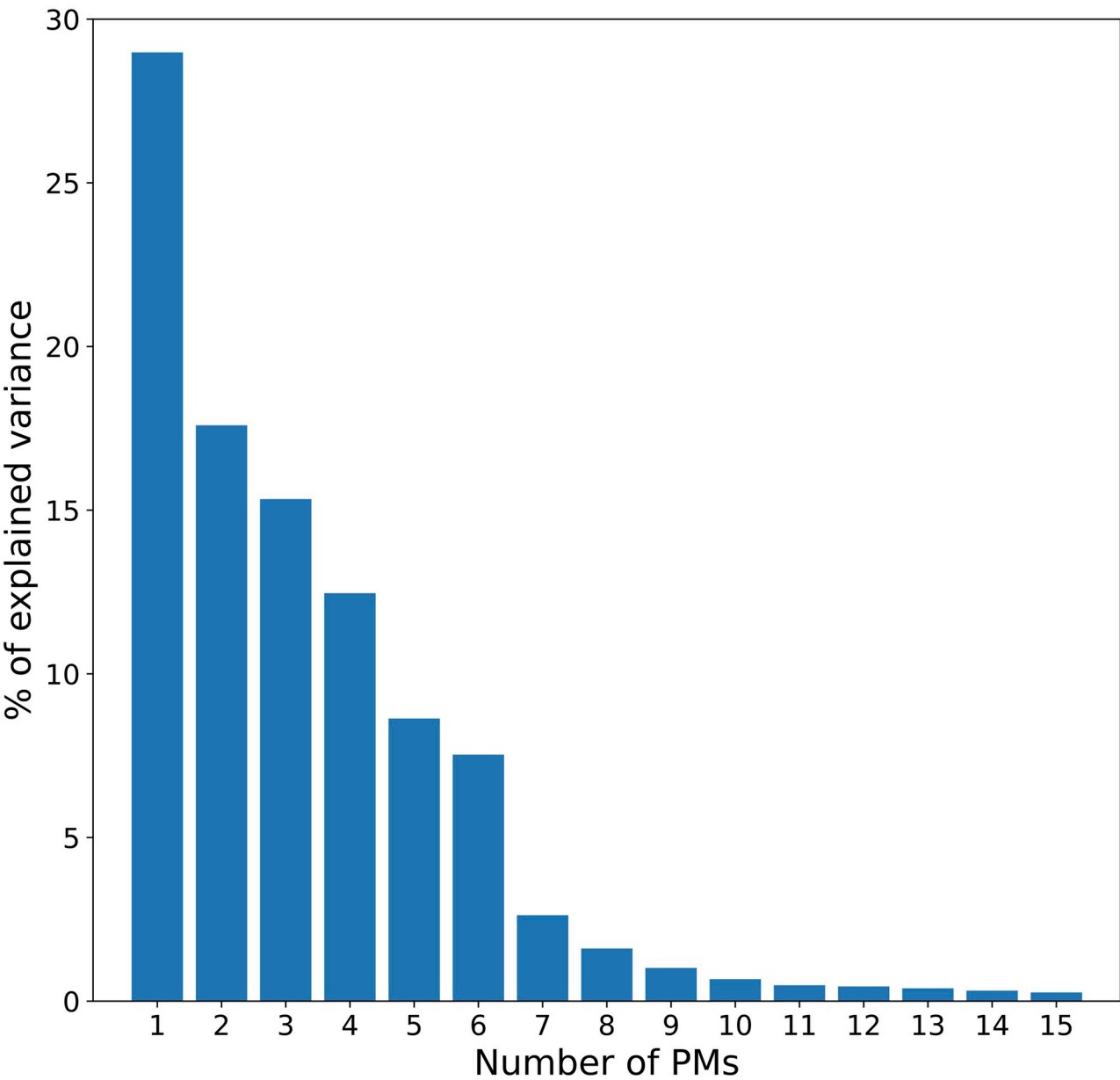

**Fig 2. Variance explained by the first 15 common PMs.**

three others), others reported lower similarity between their PMs and the common ones (e.g., Signer 6, where $sim > 0.7$ for only three PMs, and $sim < 0.5$ for four PMs).

However, most of the PMs that reported low levels of similarity when compared one-by-one displayed non-negligible similarities to multiple common PMs (e.g., PM2 to PM5 of Signer 6 or PM5 and PM6 of Signer 1). In other words, these PMs may be combinations of common PMs. This outcome suggests that, although some PMs were not perfectly similar when compared individually with common ones, the PM subspace of the signer may still account for the movements of all signers and thus reflect similar low-dimensional dimensions used to control upper-body SL motion. A demonstration of these potential combinations

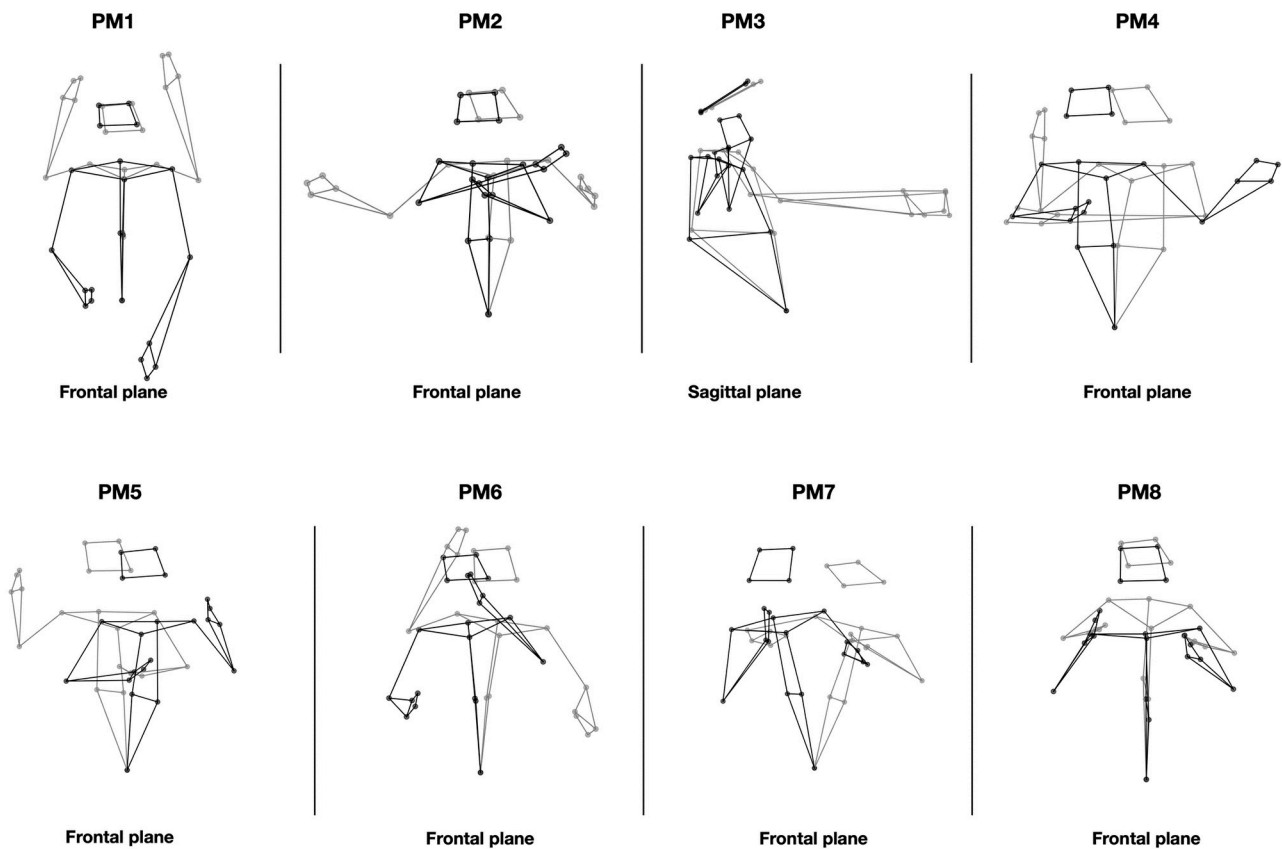

**Fig 3. The first eight common PMs.** Stick figures represent the PM at the time instants corresponding to the minimum (gray) and the maximum (black) PM weighting, across signers and examples. PMs are displayed in their main plane of motion (e.g., frontal or sagittal).

appears notably in PM1 of Signer 1 (Video 17 in S1 Dataset), which can be characterized as vertical movement of the hands (*sim* = 0.71 with common PM1), jointly with anteroposterior movement of the hands (*sim* = 0.45 with common PM3).

In summary, most of the individual PMs were similar to the common ones, although they were sometimes ranked in a different order or were combinations of several common PMs. A few PMs could eventually be interpreted as signer-specific, when the cosine similarity was significantly low and not reflecting a combination of common PMs (e.g., PM7 of Signer 5, or

**Table 1. Characterization of the first eight common PMs.** EV is the Explained Variance in original movements.

| PM | EV (%) | Description |
|---|---|---|
| 1 | 28.5 | Vertical parallel movement of the hands. |
| 2 | 17.4 | Mediolateral opposite movement of the hands. |
| 3 | 15.1 | Anteroposterior parallel movement of the hands. |
| 4 | 12.7 | Upper-body rotation around the vertical axis, jointly with parallel shift of the two hands along the mediolateral axis. |
| 5 | 8.7 | Opposite internal rotations of the arms. |
| 6 | 7.4 | Opposite vertical movement of the two hands achieved by elbow flexion. |
| 7 | 2.9 | Upper-body rotation around the vertical axis, jointly with wrists flexion. |
| 8 | 1.8 | Abduction of both shoulders while elbows are flexed, jointly with slight anteroposterior sway of the upper body. |

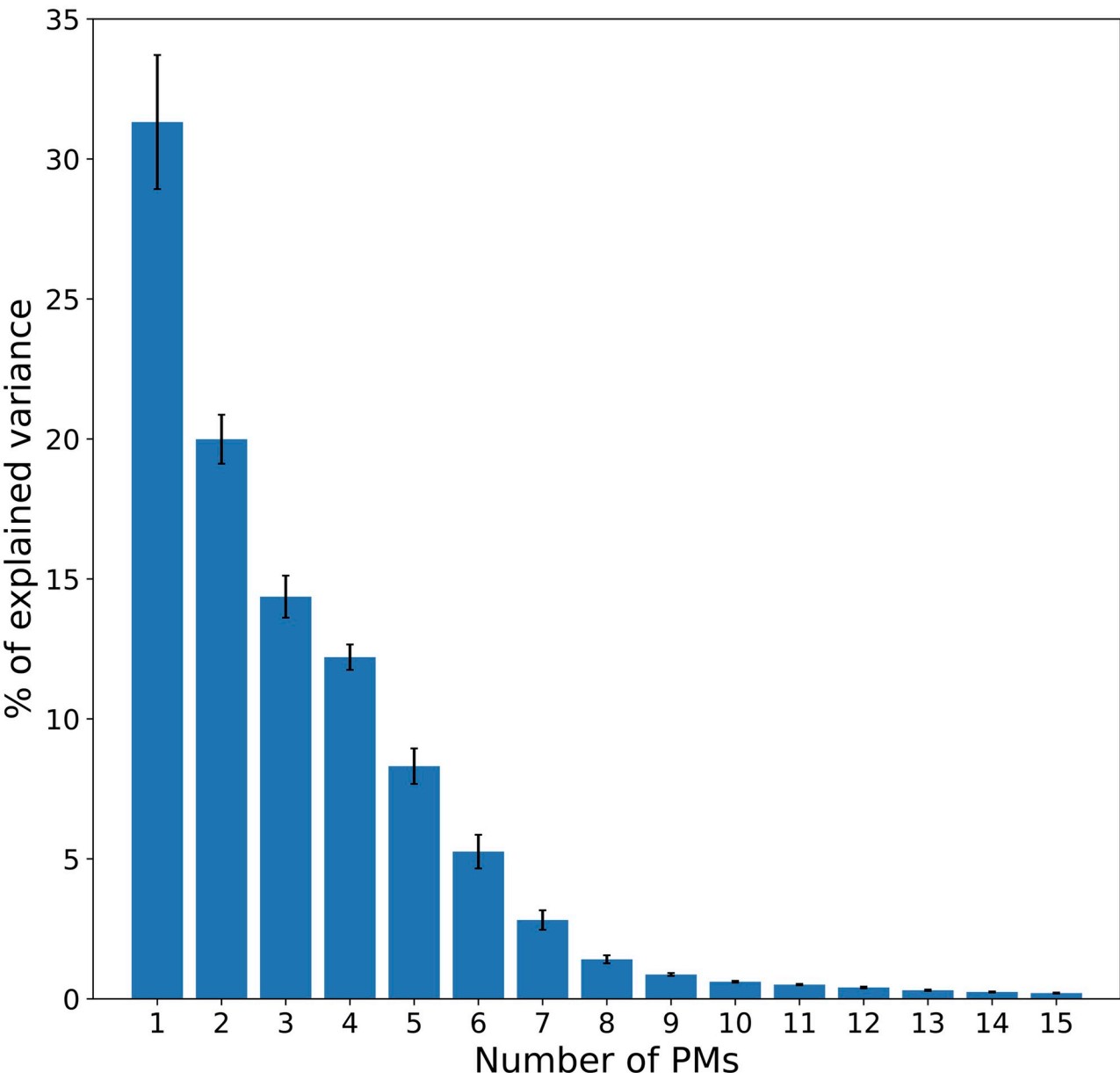

**Fig 4. Variance explained by the first 15 individual PMs.** Mean was computed across the six signers, error bars indicate standard errors across signers.

PM6 of Signer 6). Still, the reported one-by-one similarities called for further analysis investigating the extent to which the PM subspaces, rather than the PMs individually, were similar across signers. This was achieved using cross-projection similarity (i.e., for each signer, we computed the amount of variance in the movements that could be accounted for by the first N PMs of another signer). This measure revealed that the PM subspaces specific to each signer were highly similar: on average, the first eight PMs of each signer explained 96.9% of the variance in the movements of the other signers (see Fig 6).

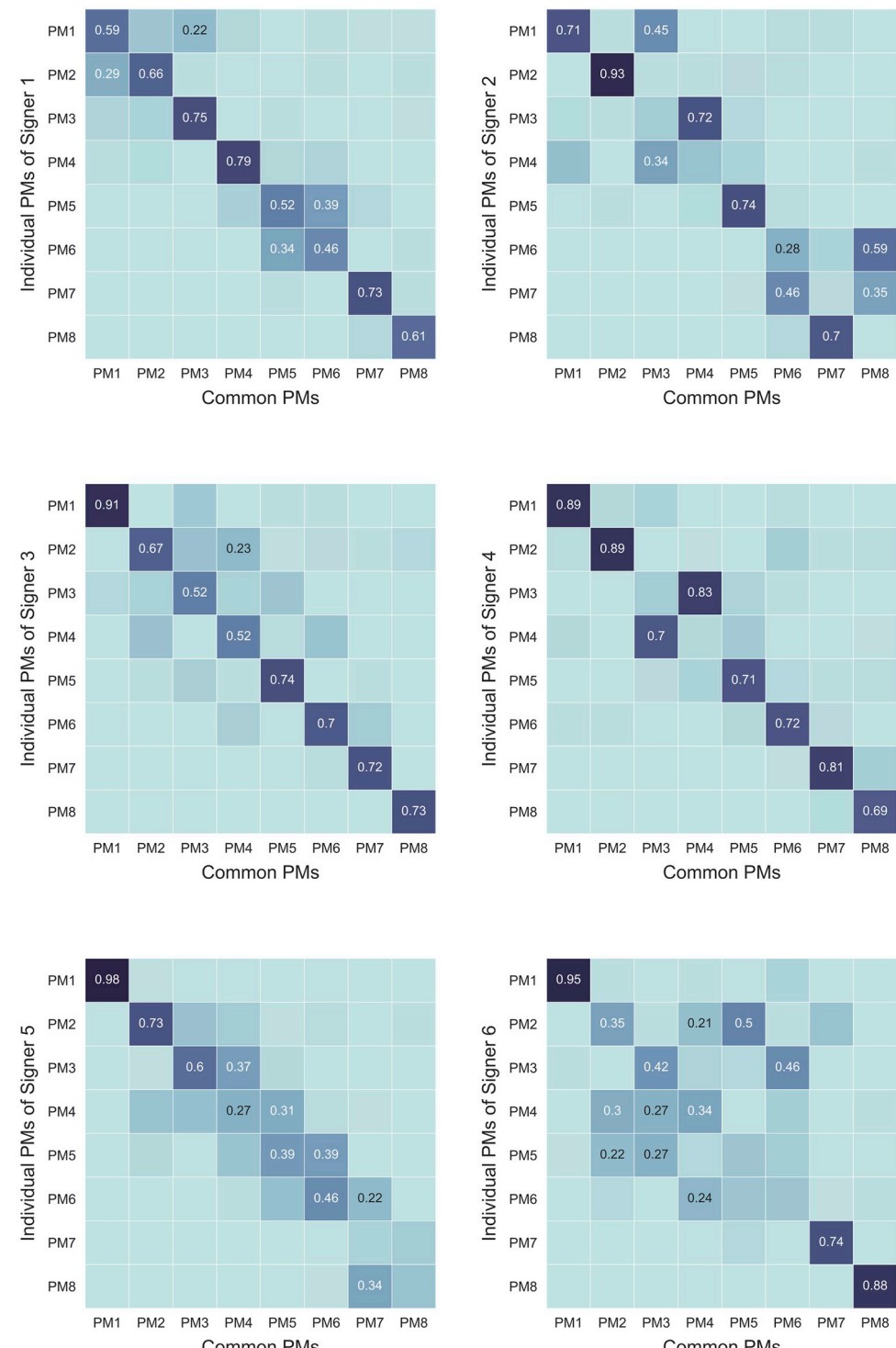

**Fig 5. Cosine similarity between individual and common PMs.** The similarity measures are specified when $sim > 0.2$, for sake of visibility.

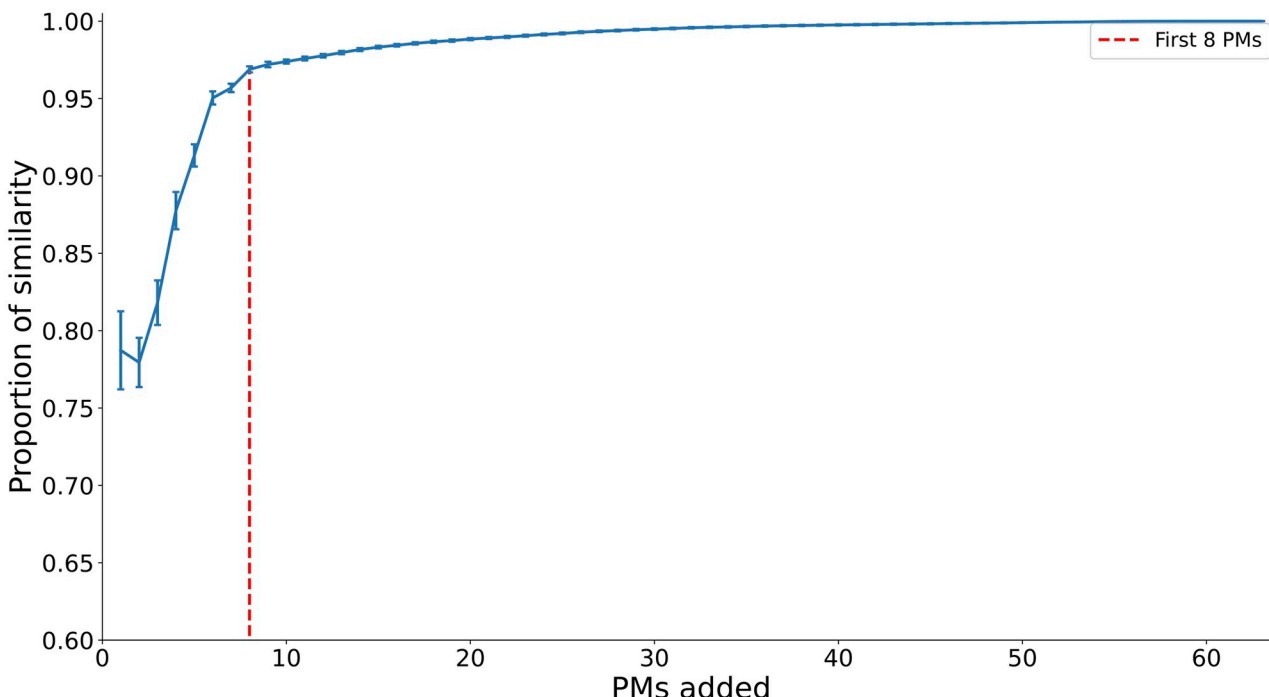

**Fig 6. Mean cross-projection similarity between PM subspaces across signers.** Mean was computed across the six signers, error bars indicate standard errors.

## Discussion

In the present paper, we used PCA to decompose the upper-body movements of spontaneous LSF into a reduced set of simpler, elementary, movements. The original motion data were transformed into a new space spanned by principal components, called principal movements (PMs). The first eight common PMs (i.e., computed on the whole dataset containing all signers) accounted for 94.6% of the variance in the movements, which suggests that the control of spontaneous SL motion can be limited to a low number of dimensions—far fewer than the multiple dozens of DOFs of the upper body—, as previously demonstrated for full-body movements, such as gait [10], diving [20], skiing [18] or juggling [21]. This outcome is also in line with prior work on the control of hand gestures, such as grasping [31, 48], piano playing [29, 49] and in particular the production of SL letters [30–33]. However, our analysis differs from these prior SL-related studies in one crucial respect. The LSF productions used in the present study are termed *spontaneous* as signers freely described pictures without any constraints in time, signs or structure. Signers could thus express themselves freely, which elicited a wider variety of SL linguistic forms (e.g., lexical signs, but also depicting signs that describe size and shapes of entities), by contrast with the highly constrained movements (i.e., ASL letters) assessed in previous studies on hand gestures. Furthermore, the present SL movements were recorded in context, within continuous discourses, which is known to give rise to further motion features and coordination properties, by contrast with isolated productions like ASL letters [34–36]. The PMs extracted from our motion dataset are thus more likely to reflect potential synergies used by signers in real-life conditions, which is crucial notably to conceive efficient real-life communication tools.

Although it is now well known that PCA is efficient in decomposing human motion into low-dimensional subspaces, the fact that it is confirmed for unconstrained continuous SL is still an intriguing result, considering the variety of analyzed movements in the discourses (e.g., gestures were neither necessarily consistent in structure nor synchronized in time across signers and examples). By contrast, all studies investigating the synergies of SL in hand gestures [30–33] were limited to a specific, well-defined, set of linguistic forms: letters of the alphabet. As precisely outlined by one of the latter studies [30], the common synergies extracted across signers producing ASL letters may be expected because of the hand postures that were targeted to conform to the same forms defined by the alphabet. Although the hand has by definition a high number of biomechanical DOFs, the production of ASL letters imposes a strong linguistic constraint on the motion, which therefore reduces the number of potentiel hand configurations (i.e., could be referred to as linguistic DOFs, as compared to biomechanical ones). Note that in this respect, our results also differ from other evidence of synergistic control strategies in full-body motion made on movements that shared similar temporal structures (e.g., gaits [10], karate's *kata* [19], dives [20] or juggling patterns [21]). Using spontaneous discourses, the present study further demonstrated that, despite a high number of biomechanical DOFs and potential linguistic gestures, upper-body movements of SL can be limited to low dimensions.

The second outcome of this study is that the low-dimensional PM subspaces were consistent across signers. First, the number of PMs needed to account for most of the variance in the movements was nearly the same for individual PMs as for common ones. Although the number of PMs retained is a simple metric, it was not trivial that it would be the same across the common and individual datasets. For instance in prior work on full-body synergies, the PMs common to all participants did not necessarily represent the original movements as well as the PMs computed for each participant [18]. For a comparison with prior SL-related work on hand gestures, a similar synergistic structure common across signers has been reported but it was more likely to occur, given that all signers were constrained by the common forms of an alphabet [30], by contrast with the free discourses used here. Given prior outcomes outlined on postural mechanisms related to gesture expertise [21, 29], we may also note that the similar number of PMs retained across the six signers of the present study could reflect that they had a similar level of expertise in SL gestures, all being fluent signers. Furthermore, although PMs of lower variance (i.e., PMs > 8) can be deemed to be directions of noise [19, 20], intriguing results including recent work on ASL hand gestures [31, 50] have shown that high-order PMs can be highly structured, which calls for further research investigating whether the high-order PMs (e.g., accounting for less than 1%-variance) of upper-body motion in spontaneous SL could be under volitional control rather than being related to noise.

Beyond the similar number of synergies, PM subspaces of the present study quantified consistent motion patterns across signers. Indeed, we showed that a high number of individual PMs (i.e. computed separately from the mocap data of each signer) were similar to common ones, although they were sometimes ranked in a different order or expressed as combinations of the common PMs. Furthermore, a cross-projection similarity analysis revealed that the first eight individual PMs specific to each signer accounted for most of the variance in the movements of all other signers. The extent to which full-body synergies are utilized across various locomotion tasks has been assessed in a few studies [46], which showed that similar PMs were recruited across tasks but that they could be prioritized differently depending on the task. As regards SL hand gestures, different signers were also reported to yield similar PM subspaces when producing ASL letters [31]. To the authors' knowledge, the present study is the first to demonstrate the similarity between synergies used across signers and across an unconstrained range of linguistic gestures.

The motion patterns described by common PMs were mostly visible in the frontal and sagittal planes, in line with the vast majority of prior studies on full-body synergies [10, 14, 15, 18, 19, 23, 24, 51], except for some PMs of juggling reported in the transverse plane [21]. In the present study, common PM4 (i.e., trunk rotation) can be described in the transverse plane but it was clearly visible from a frontal view. PM3 was best visible from a sagittal view. Still, the patterns quantified by most PMs were occurring in the frontal plane. These findings interestingly complement prior SL-related work on the synergies of hand gestures [30–33]. Indeed, the mocap data used in the present study include upper-body trajectories by contrast with its predecessors, which have investigated the kinematics of the dominant hand only. Inversely, these prior studies have applied PCA to precise recordings of finger gestures, while our mocap data were recorded on various upper limbs but only included global motion data of the wrists and hands. Both approaches provide fundamental insights into the complex structure of SL motion. Taken together, these findings could be of particular interest for improving automatic SL processing models, which requires relevant representations of both upper-body movements and finger gestures, in particular for automatic recognition [39, 52] and generation [53–56]. Further work examining larger mocap datasets and across a higher number of signers could be of interest in order to refine the definition of common PMs, which here were common to six signers only.

The successful application of PCA to the complex movements of spontaneous SL provides potential contributions to SL research for both fundamental and application purposes. First, PCA allows resynthesizing PMs in the original 3D space, which enables researchers to visualize these directions of high movement variability. For these reasons, PM decomposition has been widely used to better understand the coordinative structure of complex movements and could shed light on the motor control of SL movements, as previously investigated through common laws of motion [43, 57]. Furthermore, it allows for dimensionality reduction. This has a significant potential impact on machine learning procedures used in automatic SL tasks, in the same way as some past studies aimed to drastically lower the frame rate [58] or the number of markers [59] to reduce the bandwidth in SL telecommunication. Following the present results, the dimensionality of dense mocap datasets could be considerably reduced using only a subset of PMs while keeping most of the information. For instance, it could significantly reduce the dimensionality of the input of deep learning models used for the automatic recognition of SL and, more importantly, exploit knowledge from the structure of SL motion to improve their performance. Furthermore, both the potential to resynthesize movements from the PMs and the potential to reduce dimensionality make PM decomposition very promising for the improvement of other SL automatic tools: generation models. By shedding light on SL synergies, PCA could allow improving the existing synthesis models [53, 54]. It could also ease the incorporation of high-dimensional mocap recordings, which are known to be efficient in solving the problem of naturality and comprehensibility of virtual signers [56, 60, 61]. These potential applications to SL generation call for further work evaluating the observers comprehension of SL messages when resynthesized from a reduced set of PMs.

## Supporting information

**S1 Fig. Frequency content of the first eight common PMs.** Power Spectral Density was estimated using the Welch method.
(TIF)

**S1 Dataset. Principal movements.** Video examples of PMs shown as Point-Light Displays in frontal and sagittal planes (Total 17 files included). It includes examples of both common and individual PMs.
(ZIP)

## Author Contributions

**Conceptualization:** Félix Bigand.

**Formal analysis:** Félix Bigand, Elise Prigent, Bastien Berret, Annelies Braffort.

**Supervision:** Elise Prigent, Bastien Berret, Annelies Braffort.

**Writing – original draft:** Félix Bigand.

**Writing – review & editing:** Félix Bigand, Elise Prigent, Bastien Berret, Annelies Braffort.

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
