## [Decision Letter · Decision Letter 0]

15 Jul 2021

PONE-D-21-20070

Decomposing spontaneous sign language into elementary movements: A principal component analysis-based approach

PLOS ONE

Dear Dr. Bigand,

Thank you for submitting your manuscript to PLOS ONE. After careful consideration, we feel that it has merit but does not fully meet PLOS ONE’s publication criteria as it currently stands. Therefore, we invite you to submit a revised version of the manuscript that addresses the points raised during the review process.

Both reviewers issued severaö important concerns. Please adress them carefully.

Also, make clear, what this study actually proves.

We look forward to receiving your revised manuscript.

Kind regards,

Peter Andreas Federolf

Academic Editor

PLOS ONE

Journal Requirements:

Reviewers' comments:

Reviewer's Responses to Questions

**Comments to the Author**

1. Is the manuscript technically sound, and do the data support the conclusions?

Reviewer #1: Partly

Reviewer #2: Partly

2. Has the statistical analysis been performed appropriately and rigorously? 

Reviewer #1: No

Reviewer #2: No

3. Have the authors made all data underlying the findings in their manuscript fully available?

Reviewer #1: Yes

Reviewer #2: Yes

4. Is the manuscript presented in an intelligible fashion and written in standard English?

Reviewer #1: Yes

Reviewer #2: Yes

5. Review Comments to the Author

Reviewer #1: Dear Authors,

Thank you for submitting this manuscript and for your contribution to the movement science field overall. I have decided that 'Major Revisions' are required for the following reasons in no particular order:

-The main focus and finding of this study, that sign language movement is constrained to a low dimensional space isnt a novel finding (https://doi.org/10.1038/s41467-020-17404-0) but what is interesting is that it replicates this finding with a more liberal dataset where participants were more self-directed. As you will see in this paper also, PCA was applied successfully, making the main findings here rather redundant. An effort should be made to build on previous work or make a direct connection to related work and how it has been supported or not.

- The variance accounted for as a metric of complexity is not very comparable across datasets as it is sensitive to various non-movement related differences including noise, number of markers included etc. Therefore, premising the study on this finding is valid when making comparisons within the same dataset but is not advised across datasets. There is no justification explicitly provided for the number of synergies extracted. Your results are as one would expect, that sign language movements are one of the more complex of movements but using skiing as a comparison here isnt very informative. There are a number of hand gesture related studies in the literature that may make for more insightful comparisons for the underlying motor control mechanisms. Linking the findings back to how they may help with the greater goal you have mentioned (technology development) throughout the paper will drive home the point of the study also.

-Although higher order PMk can contain subtle movement characteristics as you have shown, these are also vulnerable to noise and so it may be worth including in your study an analysis of the signal strength in these synergies. If you would like to say that these synergies were consistent across participants, and wouldnt drastically change if a participant was added/left out, this approach is advisable, and would make the results generalizable. One method for this has been conducted in the research you cite, a leave one out cross validation (10.3389/fnagi.2018.00022).

-Although you aim for a qualitative analysis, this may be difficult when properly addressing your aims and describing the results of a computational method while adding to the existing knowledge base. The capacity for the PCA protocol to capture individual differences and common synergies is well known, and the amount of words spent on this is not justified based on the existing evidence base. The similarities and differences between individuals is indicated using discrete examples but not quantified, and goes against a motivation of the study which was to holistically quantify movement parameters. Perhaps illustrating some metrics for how this approach has holistically captured the movements rather than discrete examples would be more informative. Again, linking these findings to those in the literature related to sign language is advisable. Although you aim for a qualitative analysis, perhaps an exploratory approach that involves a mixture of qualitative and quantitative findings would be more suitable for addressing the aims of the study. The following paper may be of use for such an approach: doi: 10.3389/fspor.2020.596063. Much of this analysis could have been conducted using the PManalyzer software (10.3389/fninf.2019.00024), if this was indeed used, please cite.

- Please proof read the manuscript for spelling and grammar and reference using a single format throughout the paper.

Overall the study has the potential for publication if the aims of the paper are adequately addressed, the methodology and results are properly justified and generalizable, comparisons are made within context and a consistent message is provided throughout the paper of novel results.

I look forward to reading your revised version.

Reviewer #2: Summary:

The manuscript „Decomposing spontaneous sign language into elementary movements: a principal component analysis-based approach” describes how the French sign language of six participants can be decomposed into principal movements (PMs) using motion capturing data and PCA as decomposing method. One finding was that the first eight PMs, which were calculated considering the data of all six participants, reflected 94.9% of the overall variance. Moreover, the first eight PMs calculated individually for every participant reflected 95.9% (SD = 0.6%). Since the movements of the different participants were not synchronized, the authors interpreted this similarity by stating that PCA is a valid method to decompose sign language. Moreover and following the argument of the authors, sign language appears to follow dominant movement patterns which are similar upon different signers. The manuscript is well written besides some small misspellings. However, there are some major and minor concerns which I would like to address below.

Major Concerns:

Abstract: The last sentence says that the results suggest “that SL may have a common structure…”. Isn’t that obvious since it is supposed to be understood by different people who potentially never met each other, but who use the same technique to communicate? I miss a statement about the importance and relevance of the findings. What does this finding indicate or which implications could be formulated based on the results? Was the only purpose of the study to validate if PCA is a method to decompose sign language? If yes, please clarify why this is of importance.

Methods: Did the participants write an informed consent and/or was the procedure approved by an Ethics committee?

Line 113: The authors state that normalization took place by subtracting the participants average posture from the posture vectors, which is more a centering of the data and does not ensure completely that each participant contributes the same variance to the newly created matrix (see reference 21). I would suggest to also normalize the vectors afterwards by dividing their mean Euclidean distance, also described in reference 20 and 21. Especially for the PCA calculated for all six participants, this would be crucial to only “extract common PMs” as stated in line 141.

Were the markers weighted based on their relative body mass, e.g. as defined by de Leva?

Line 151: No statistical analysis was conducted which in my opinion makes it difficult to objectively evaluate the findings of the study. Yes, taking a look at the visualizations or the figures, the PMs appear to be quite similar. However, to state that the “PMs describe highly similar motion patterns” (line 283) is a very subjective statement in my opinion and should be tested using statistics. E.g. tests for correlation could be applied on the postures during the PMs as shown in Figure 4.

Line 283: Following my comment stated before, to me it is not possible to make such a statement based on the results presented in the manuscript as they are just based on descriptive evaluation.

Minor Concerns:

Line 11: The authors mention the importance of hands, torso and face for SL. I see that it is difficult to evaluate the mimic in the face and that it is not the focus of their study. However, why was the amount of markers reduced from 27 to 19 (line 92)? That takes away the chance to take the torso movement into account.

Line 41: Misspelling: “Egeinvoices”

Line 58-61: PCA was also already used to search for and find changes in postural control due to impairments like cerebral palsy (Rethwilm, R., Böhm, H., Dussa, C. U. & Federolf, P. (2019). Excessive lateral trunk lean in patients with cerebral palsy: is it based on a kinematic compensatory mechanism?. Frontiers in bioengineering and biotechnology, 7, 345.) or after head shaking (Wachholz, F., Kockum, T., Haid, T. & Federolf, P. (2019). Changed temporal structure of neuromuscular control, rather than changed intersegment coordination, explains altered stabilographic regularity after a moderate perturbation of the postural control system. Entropy, 21(6), 614.). Two articles that might fit in this context as well, as being deaf can be related to a neurological impairment, too.

Line 85: I miss some descriptive data about the participants here. I found the average age in reference 26, but could not find any information about the sex, about the inclusion criteria and, if existent, the exclusion criteria.

Moreover, is it valid to talk about “common” PMs, when only six participants were assessed?

Line 98: Which program was used to calculate the PCA? Was it a custom-made code or an already existing one?

Line 121: Misspelling: “Egeinvectors”

Line 178: I mentioned it in the major concerns already, but again: Did the authors perform any statistical test to validate their descriptive interpretation?

Line 264: I don’t think that the comparison leading to this result is valid. Skiing is a movement which involves a large amount of joints and therefore results in more degrees of freedom. This ultimately results in a higher number of PMs, which are needed to describe the movement compared to SL – or at least to SL analyzed in the current manuscript – as only the upper body was considered.

General comment: Thank you for the animated visualizations, they help to get a better understanding of the PMs. One suggestion is to implement the name of the PM that is visible and the participant number into the video or at least name the video-file which shows PM1, e.g. ”pm1.mp4”. I assume that “video1” shows PM1 but to avoid misunderstanding I would propose to name the files accordingly to their PM.

6. PLOS authors have the option to publish the peer review history of their article (what does this mean?). If published, this will include your full peer review and any attached files.

Reviewer #1: **Yes: **David Ó'Reilly

Reviewer #2: No

---

## [Author Response · Author response to Decision Letter 0]

18 Sep 2021

See separate pdf file "Response to Reviewers"

---

## [Decision Letter · Decision Letter 1]

20 Oct 2021

Decomposing spontaneous sign language into elementary movements: A principal component analysis-based approach

PONE-D-21-20070R1

Dear Dr. Bigand,

We’re pleased to inform you that your manuscript has been judged scientifically suitable for publication and will be formally accepted for publication once it meets all outstanding technical requirements.

Kind regards,

Peter Andreas Federolf

Academic Editor

PLOS ONE

Reviewers' comments:

Reviewer's Responses to Questions

**Comments to the Author**

1. If the authors have adequately addressed your comments raised in a previous round of review and you feel that this manuscript is now acceptable for publication, you may indicate that here to bypass the “Comments to the Author” section, enter your conflict of interest statement in the “Confidential to Editor” section, and submit your "Accept" recommendation.

Reviewer #1: All comments have been addressed

Reviewer #2: All comments have been addressed

2. Is the manuscript technically sound, and do the data support the conclusions?

Reviewer #1: Yes

Reviewer #2: Yes

3. Has the statistical analysis been performed appropriately and rigorously? 

Reviewer #1: Yes

Reviewer #2: Yes

4. Have the authors made all data underlying the findings in their manuscript fully available?

Reviewer #1: Yes

Reviewer #2: Yes

5. Is the manuscript presented in an intelligible fashion and written in standard English?

Reviewer #1: Yes

Reviewer #2: Yes

6. Review Comments to the Author

Reviewer #1: All comments were satisfactorily addressed and the newest manuscript version is much improved. I am therefore happy to recommend this study for publication.

Reviewer #2: Congratulations to the author's, as they managed to significantly improve the manuscript, which in my opinion is now suitable for publication.

7. PLOS authors have the option to publish the peer review history of their article (what does this mean?). If published, this will include your full peer review and any attached files.

Reviewer #1: **Yes: **David Ó' Reilly

Reviewer #2: **Yes: **Felix Wachholz

---

## [Editor Report · Acceptance letter]

22 Oct 2021

PONE-D-21-20070R1 

Decomposing spontaneous sign language into elementary movements: A principal component analysis-based approach 

Dear Dr. Bigand:

I'm pleased to inform you that your manuscript has been deemed suitable for publication in PLOS ONE. Congratulations! Your manuscript is now with our production department. 

Kind regards, 

on behalf of

Dr. Peter Andreas Federolf 

Academic Editor

PLOS ONE